# Melatonin and TGF-β-Mediated Release of Extracellular Vesicles

**DOI:** 10.3390/metabo13040575

**Published:** 2023-04-18

**Authors:** Klaudia Piekarska, Klaudia Bonowicz, Alina Grzanka, Łukasz M. Jaworski, Russel J. Reiter, Andrzej T. Slominski, Kerstin Steinbrink, Konrad Kleszczyński, Maciej Gagat

**Affiliations:** 1Department of Histology and Embryology, Collegium Medicum in Bydgoszcz, Nicolaus Copernicus University in Torun, 85-092 Bydgoszcz, Poland; klaudia.mikolajczyk@cm.umk.pl (K.P.); klaudia.bonowicz@cm.umk.pl (K.B.); agrzanka@cm.umk.pl (A.G.); lukaszmjaworski@gmail.com (Ł.M.J.); 2Department of Cell Systems and Anatomy, UT Health, Long School of Medicine, San Antonio, TX 78229, USA; reiter@uthscsa.edu; 3Department of Dermatology, Comprehensive Cancer Center, University of Alabama at Birmingham, Birmingham, AL 35294, USA; aslominski@uabmc.edu; 4Pathology and Laboratory Medicine Service, VA Medical Center, Birmingham, AL 35294, USA; 5Department of Dermatology, University of Münster, Von-Esmarch-Str. 58, 48149 Münster, Germany; kerstin.steinbrink@ukmuenster.de

**Keywords:** melatonin, transforming growth factor β, extracellular vesicles, cell-to-cell communication

## Abstract

The immune system, unlike other systems, must be flexible and able to “adapt” to fully cope with lurking dangers. The transition from intracorporeal balance to homeostasis disruption is associated with activation of inflammatory signaling pathways, which causes modulation of the immunology response. Chemotactic cytokines, signaling molecules, and extracellular vesicles act as critical mediators of inflammation and participate in intercellular communication, conditioning the immune system’s proper response. Among the well-known cytokines allowing for the development and proper functioning of the immune system by mediating cell survival and cell-death-inducing signaling, the tumor necrosis factor α (TNF-α) and transforming growth factor β (TGF-β) are noteworthy. The high bloodstream concentration of those pleiotropic cytokines can be characterized by anti- and pro-inflammatory activity, considering the powerful anti-inflammatory and anti-oxidative stress capabilities of TGF-β known from the literature. Together with the chemokines, the immune system response is also influenced by biologically active chemicals, such as melatonin. The enhanced cellular communication shows the relationship between the TGF-β signaling pathway and the extracellular vesicles (EVs) secreted under the influence of melatonin. This review outlines the findings on melatonin activity on TGF-β-dependent inflammatory response regulation in cell-to-cell communication leading to secretion of the different EV populations.

## 1. Introduction

The proper functioning of the cells that build the vessel walls is the basic condition for maintaining homeostasis in the body [1,2,3]. In the heart, arteries, capillaries, and veins the multi-functional nature of the ECs relies on providing an anti-inflammatory and anti-coagulatory surface in the physiological state for the remaining cells [4,5]. On the other hand, the vessel wall layer controls the adhesion and migration of inflammatory cells under imbalanced conditions. Any disturbance that causes the disruption of intercellular connections of ECs and vessel unsealing may lead to leakage of immune cells from the lumen to adjacent tissues, and initiation of inflammation [6,7,8]. Typically, this process is part of the innate immunity and physiological response to injury; however, if prolonged, it constitutes a major factor in the development and complications of atherosclerotic cardiovascular diseases. For this reason, anti-inflammatory therapies involving the stabilization of chemotactic cytokines are the current trend in cardiovascular medicine [9,10,11].

Chemotactic cytokines, also known as chemokines, are a group of proteins that stimulate the movement of leukocytes and control their migration from the blood to tissues [12]. This property determines their undeniable role in the formation of an inflammatory focus. The altered concentration of chemokines in individual disease states may be the target of research—as potential diagnostic or prognostic markers, as well as a promising target for therapeutic interventions [13,14].

At initiation sites of inflammation, chemokines direct the progression of the immune response based on the leukocyte migration across the endothelium [15,16]. The inflammatory reaction is a multi-stage process controlled by the interaction of adhesion molecules located on the luminal surface of endothelial cells with surface leukocyte receptors [17,18]. Chemokines mobilize the immune system cells to concentrate at the focus of inflammation and maintain homeostasis of the body. This process is referred to as extravasation, which involves a cascade of reactions where the first step is the contact of leukocytes with the EC layer, which is called leukocyte rolling [19]. During slow rolling, leukocytes can interact with the chemoattractant present on the surface of the endothelium, which binds to specific transmembrane receptors linked to intracellular Gi proteins [20]. Signals transmitted by this class of receptors increase the affinity of integrins, which ensures the stable adhesion of leukocytes to endothelial cells. Then, integrins can bind to adhesive proteins, e.g., intercellular adhesion molecule 1 (ICAM-1), and vascular cell adhesion molecule 1 (VCAM-1) [21,22]. The next step is cytokine-dependent activation and selectin-dependent tight adhesion, which consequently allows cells to pass through the endothelial layer to the surrounding tissues by diapedesis [23]. The first stages depend mainly on selectins, including *E*- and *P*-selectins, which alternately bind briefly and release from bonds to carbohydrate groups, slowing down the movement of leukocytes in the vessel [24]. Their expression is regulated by cytokines, while their ligands are expressed on specific leukocyte subpopulations [25,26,27]. The expression of selectins and selectin ligands is limited to the microvilli present on the surface of leukocytes, allowing for effective interaction with vascular ECs. The chemokine activity is therefore essential for the initiation and course of a proper immune system response and regaining internal balance [12,28]. Chemokines play an extremely important role in the development of cardiovascular diseases, i.e., the progression of atherosclerotic plaque [29,30]. The initial stages of atherogenesis are associated with the exposure of the CXC chemokine ligand (CXCL) by ECs, which are regulated by lysophosphatidic acid, a component of low-density lipoproteins (LDL) [31]. For example, the chemokine CXCL1 may recruit leukocytes to infiltrate the vascular wall and influence the progression of atherosclerosis in response to stimulation by phosphatidic acid (PA) [32,33]. On the other hand, CCL17 inhibits the influence of regulatory T cells in the promotion of atherosclerotic lesions [34,35]. The expression of the CXCL12 chemokine in endothelial cells, which stabilizes atherosclerotic plaques, can be induced by microribonucleic acid (miR)-126 [36,37]. Another chemokine receptor, CX3CR1, is responsible for sending strong signals that prolong the survival of monocytes and macrophages, which protects them from apoptosis. In contrast, CXCL5 reduces the formation of foam cells from macrophages [38,39,40,41]. Thus, the cytokine essential for many key cellular processes and for maintaining the homeostasis of every cell in the body, TGF-β, has been considered [42,43]. Despite many studies, its action is still difficult to characterize, due to its pleiotropic properties. It has been well-explained in cancer research and has been referred to as “the TGF-β paradox”. However, in cardiovascular medicine, the role of TGF-β is still ambiguous. On the one hand, its protective role is emphasized, and it is considered a major driver of vascular inflammation [44,45]. To date, misregulated TGF-β signaling in humans has been linked to the onset of vascular pathologies and cardiovascular diseases such as arteriovenous malformations (AVMs), aneurysms, atherosclerosis, cardiac fibrosis, vascular remodeling of the retina (retinopathy), and valvular heart disease [46,47].

Growing evidence suggests that melatonin synthesized in pinealocytes exerts protective effects against atherosclerosis-based vascular diseases, but these mechanisms are poorly understood [48,49]. Melatonin possesses anti-inflammatory capacities with benefits in protecting the structural and functional integrity of vascular endothelium against aging-, oxidative-stress-, lipopolysaccharide-, and ischemia-induced damage [50,51,52]. These profound effects are mainly exacerbated due to its antioxidant properties affecting the reduction of reactive oxygen species (ROS), which are the driving force of vascular pathology [53]. Despite some contradictions, most of the data claims that melatonin is a promising supplement that has no side effects [54]. Herein, we summarize the most established benefits of melatonin in the vascular system, focusing on the molecular mechanisms regulating the TGF-β signaling pathway. 

The TGF-β signaling mechanism can modify the extracellular vesicle (EV) secretion process, the evidence for which points to an important connection of EVs with inflammatory response biology [55]. EVs form a heterogeneous group of nanoparticles, providing an extremely important means of transmitting information between cells, without direct contact [56,57]. Recently, the intensity of research focused on EVs has significantly increased, paying particular attention to their activity in intercellular communication, for which bioactive molecules carried by vesicles between cells are responsible. The activity of the TGF-β signaling pathway in the course of the inflammatory response may regulate the secretion of membrane structures in order to modulate intercellular communication, allowing for the restoration of intracorporeal homeostasis [58,59]. Particularly interesting seems to be the currently little-known effect of melatonin on the cellular environment. The presence of this neuromolecule not only modulates the inflammatory response, but also affects the biogenesis, EV secretion amount, and composition of membrane vesicle cargo [60,61].

The purpose of this review is to provide a detailed description of the EV secretion dependent on the TGF-β signaling pathway mediated by melatonin. We focus on the molecular cargo and EVs’ association with disease and emerging strategies for their therapeutic exploitation.

## 2. Development and Progression of Vessel Wall Inflammation

The inflammation linked to the onset of atherosclerosis occurs between the layers of large and medium arteries, more specifically in the subendothelial space [62,63]. The endothelium is the innermost part of the blood vessels (arteries, veins, and capillaries) and consists of a single, semi-permeable layer of cells that is constantly regulated by local hemodynamic forces [64,65]. Areas of low endothelial shear stress (ESS) are the most common predictor of atherosclerotic plaque formation [66]. Low ESS, tangential stress due to the friction of the flowing blood on the endothelial surface, is also considered a focal pro-inflammatory stimulus, which contributes to endothelial dysfunction [67,68]. Another crucial factor important for maintaining endothelial homeostasis is the balance between vasodilation and vasoconstriction, mainly mediated by endothelium-derived nitric oxide (NO) bioavailability and other relaxing and contracting factors, such as angiotensin, endothelin-1 (ET-1) and oxidants [69,70]. NO production is highly dependent on the activity of the endothelial NO synthase (eNOS), also influenced by shear stress force on mechanoreceptors [71,72]. Therefore, oxidative stress-induced endothelial dysfunction, in terms of vasomotor disturbances, is the earliest event in atherogenesis, quickly followed by tissue repair mechanisms [73,74].

Disabled endothelium is leaky, adhesive, and unable to relax vascular smooth muscle cells. The disruption in the normal function of the endothelial cells is inseparably accompanied by a gradual infiltration of immune cells [75]. Simultaneously, released reactive oxygen species (ROS) induce the nuclear factor kappa-light-chain-enhancer of activated B cell (NF-κB) expression, which culminates in the increase in the expression of cytokines involved in further ROS production [76,77]. TNF-α is a key cytokine that inhibits endothelium-dependent nitric oxide (NO)-mediated vasorelaxation by activating sphingomyelinase, resulting in ^•^O_2_^−^ production in the ECs [78,79,80]. TNF-α is also a potent pro-inflammatory cytokine, which promotes inflammatory endothelial activation by upregulating the expression of VCAM-1 and ICAM-1, allowing lymphocyte and monocyte adhesion [81]. The monocytes then transmigrate to the subintimal space through the interaction of monocyte chemotactic protein-1 (MCP-1) with the CCR2 receptor, where they differentiate into macrophages [82,83]. A particularly important process for plaque formation is the internalization of cholesterol-rich oxidized lipoproteins by monocytes, giving them a foamy appearance and secreting local cytokines, as well as ROS [84,85,86]. Other types of immune cells, such as DCs, T cells, B cells, and neutrophils participate in intraplaque inflammation [87,88]. The perpetuation of pro-inflammatory and oxidative atherosclerotic stimuli results in the recruitment of more macrophages, mast cells, and activated T and B lymphocytes, which enhance vascular lesions, which in turn release cytokines (i.e., interleukin-1*β* (IL-1*β*), TNF-α), increase the leukocyte extravasation to the submembrane space and maintain chronic inflammation [89,90].

The artery wall structure also consists almost entirely of circumferentially oriented vascular smooth muscle cells (VSMC), surrounding the ECs and constituting the tunica media. The VSMC is involved in the crosstalk between immune cells and ECs during all stages of atherosclerosis [91,92]. ECs-derived relaxants such as NO lower the activity tone of VSMCs, leading to (flow-mediated) vessel dilation to counteract the initial increase in wall shear stress and contribute to pathological vascular remodeling [93,94,95].

## 3. The TGF-β Signaling Pathway in the Cardiovascular System

TGF-β is one of the crucial mediators in the pathophysiology of cardiovascular diseases such as atherosclerosis and abdominal aortic aneurysm (IAA) [96,97]. This highly complex polypeptide growth factor is also described as a multifunctional cytokine that elicits its effects in the vascular system via an influence on endothelial cells, smooth muscle cells, and regulation of extracellular matrix (ECM) deposition [98,99]. TGF-β family member proteins are involved in a large variety of cellular processes, including the induction of proliferation, apoptosis, migration, adhesion, ECM protein production, and cytoskeletal organization [100,101].

The perturbations in TGF-β signaling are linked to vascular-wall inflammation, thickening, and remodeling. The most abundant isoform of the family in the cardiovascular system is TGF-β1, present in ECs and VSMC populations, but also in the myofibroblasts, macrophages, and other hematopoietic cells. The outcome of cellular response to TGF-β depends on the signaling mechanisms regulated both extracellularly and intracellularly [102,103]. TGF-β is produced in an inactive form and stored in the ECM as part of a large latent complex (LLC) consisting of TGF-β, latency-associated peptide (LAP), and latent TGF-β binding protein (LTBP) [104,105,106,107]. The newly synthesized TGF-β binds to the pro-domain, called LAP propeptide via covalent and non-covalent linkage and forms a small latent complex (SLC) to keep the molecule in a biologically inactive state and to maintain a conformation suitable for dimerization [108,109]. LTBP connects with SLC through covalent bonding and targets and stabilizes LLC in ECM rich in fibrillin and fibronectin. The latent TGF-β activation process is dependent on the cell context and may result from a proteolytic cleavage within the LAP pro-domain, which can be stimulated by factors such as plasmin, cathepsin, matrix, and metalloproteinases and the subsequent release of the mature TGF-β and/or a conformational change in the LAP, allowing exposure of the TGF-β ligand [104,105,106,107,108,109]. Bioactive ligands and unmasked sites of TGF-β bind to a TGF-β type II receptor (TGF-βRII), also referred to as activin receptor-like kinases (ALKs) at the cell surface [110]. The activated TGF-βRII then recruits and activates the TGF-β type I receptor (TGF-βRI) by trans-phosphorylation [111]. TGF-β cellular responses are also regulated by TGF-βRIII (also termed β-glycan), which exhibits no enzymatic activity but is considered an important helper molecule that presents TGF-β to TGF-βRII and facilitates its binding [112,113]. In the canonical TGF-β signaling pathway, trans-phosphorylation of TGF-βRI induces phosphorylation of transcriptional effector proteins, receptor-activated small mothers against decapentaplegic (R-Smads) such as Smad2 and Smad3 [114,115,116].

In endothelial cells, low TGF-β concentrations in ECs can activate the Smad1/5/8-based pathway. The Smads classification also includes inhibitory Smads (I-Smads, Smad6/7). Upon phosphorylation, R-Smads associate with Smad4 (Co-Smad), enter the nucleus, and regulate the transcription of TGF-β responsive genes [117,118,119]. The Smad-independent pathways are also important for the response to TGF-β stimulation, and include the Ras homologous (Rho) protein family, Src homology 2 domain-containing transforming protein 1 (ShcA), Ras-related C3 botulinum toxin substrate (RAC), rat sarcoma virus (RAS) protein family, cell division control protein 42 homologs (CDC42), TNF-α receptor-associated factor 6 (TRAF6), phosphoinositide 3-kinase (PI3K), transforming growth factor beta-activated kinase 1 (TAK1), partitioning-defective protein 6 (PAR6), mitogen-activated protein kinase 1 (MAP3K1), protein phosphatase 2 (PP2A) and death-associated protein 6 (DAXX) [120,121,122,123,124,125].

## 4. Effect of Melatonin on the TGF-β Signaling

The elevated expression level of TGF-β1 mRNA is observed during the development and progression of a variety of vascular diseases, including coronary artery disease (CAD) [126,127]. The cellular response to the TGF-β1 stimulation also depends on its proper synthesis, secretion, and activation. TGF-β’s effect on blood vessel function is concentration-dependent. The pleiotropic actions of this cytokine on the ECs depend mainly on factors such as EC origin, serum composition, cell density, and the combination of TGF-β receptors expressed on the cell surface [128,129,130]. In vitro studies on HCAEC confirm that TGF-β1 overexpression significantly promotes apoptosis, while TGF-β1 siRNA significantly inhibits cell apoptosis [131]. Moreover, activation of endothelial TGF-β signaling is one of the primary drivers of atherosclerosis-associated vascular inflammation, contributing to endothelial activation and increased vascular permeability [132]. EC treatment with TGF-β induces the expression of a number of pro-inflammatory cytokines, chemokines and their receptors (including CCL2), leucocyte adhesion molecules including ICAM-1 and VCAM-1, and matrix metalloproteinases (MMP2) as well as fibronectin, a pro-inflammatory ECM component long linked to inflammation [133,134]. Based on these data, the inhibition of TGF-β1 expression may serve as a target for the treatment of different types of cardiovascular diseases [135,136].

Recent reports indicate that one of the potent inhibitors of TGF-β signaling is melatonin [137]. Melatonin, structurally determined as 5-methoxy *N*-acetyl tryptamine is an indoleamine nocturnally released by the pineal gland into the blood and cerebrospinal fluid [138]. The melatonin secretion mechanism has been fixed by the endogenous circadian rhythm generator, which is connected with the pineal gland in the suprachiasmatic nucleus (SCN) localized into the anterior hypothalamus [139,140]. Information about the lighting conditions of the environment reaches the pineal gland through a complex neural pathway starting in the retina and covers the following signaling itinerancy: retina → retino-hypothalamic tract → SCN → paraventricular nucleus → medial forebrain bundle → tectum diencephalon → intermediate-lateral nucleus of the spinal cord → superior cervical ganglion → postganglionic sympathetic fibers → pineal pinealocytes [141]. Tryptophan has been defined as the initial compound for the production of melatonin, which after hydroxylation and decarboxylation is converted into serotonin. The transformation of this chemical compound to melatonin is based on the activity of two crucial enzymes for the entire process. The first is *N*-acetyl-transferase (NAT), which catalyzes the serotonin *N*-acetylation, whereas hydroxy indole-*O*-methyltransferase (HIOMT) carries out o-methylation, leading directly to the formation of melatonin. The melatonin lipophilic structure determines its pleiotropic properties, which allow it to pass through all biological barriers in the body [142,143]. The hydrophobic structure also determines the possibility of interacting with several biochemical pathways and indirectly and directly affecting other tissues and cells. Due to its lipophilicity, melatonin concentrates in membranes including those of mitochondria, and in the cell’s nucleus [144]. The melatonin presence in the mitochondria is strongly associated with the participation of this hormone in the body’s immune reactions associated with disorders of homeostasis caused by oxidative stress [145]. This condition consists in disturbing the balance between the by-products of metabolic changes, i.e., reactive oxygen species (ROS), and the ability to remove them from the body [146]. Many publications report on the ability of melatonin to capture free radicals, thus protecting cells from their harmful effects. Melatonin enhances the activity of antioxidant enzymes, affecting the redox potential in various types of cells. Melatonin scavenges free radicals to form kynuramine compounds such as cyclic 3-hydroxymelatonin (C3-OHM) and *N*^1^-acetyl-5-methoxykynuramine (AMK), but also *N*^1^-acetyl-*N*^2^-formyl-5-methoxykynuramine (AFMK). As mentioned, melatonin can modulate the cell membranes’ redox potential by increasing antioxidant cellular defense, either enzymatic or non-enzymatic, but also by protecting key redox proteins such as thioredoxin 1 (Trx1) from the oxidative mechanism. It is a cascade reaction pathway, independent of the presence of receptors on the surface of other cells, which leads to reductions in the free radicals’ deleterious effects [147,148].

However, for the most part, immunoregulatory effects of melatonin are based on the interaction with membrane and nuclear receptors located in the central nervous system (CNS), eyesight organs, skin, digestive tract, liver, heart, arteries, kidneys, prostate gland, and uterus [149]. The mechanism of melatonin action by binding to membrane receptors is based on the reduction in cyclic adenosine monophosphate (cAMP) concentration, which affects the signaling pathways of a number of biological signals’ secondary transmitters. The significant engagement of cAMP, inositol trisphosphate (IP3), cyclic guanosine monophosphate (cGMP), diacylglycerol (DAG), or arachidonic acid leads to changing patterns of enzyme activities [150]. In addition, melatonin is involved in the transmission of information based on the release of calcium into the cytosol by stimulating the activity of phospholipase C, which catalyzes the hydrolysis process. As a result of this process, among others, IP3 is formed, passing to the plasma reticulum, where Ca^2+^ ions are stored, strongly stimulating the increased secretion of these ions [151]. Melatonin has an affinity for orphan nuclear receptors—retinoid orphan receptors/retinoid Z receptors. The activity of nuclear receptors particularly affects leukocytes, by inhibiting the action of 5-lipoxygenase, the enzyme responsible for cellular leukotriene biosynthesis from arachidonic acid, underlying inflammatory processes [152]. Another mechanism of melatonin action is based on binding to intracellular proteins, such as calmodulin, calreticulin, and tubulin, but its antioxidant properties also promote the creation of a melatonin-dependent antioxidant system [153,154].

Although melatonin plays a significant role in maintaining homeostasis and protecting tissue functional activity under exposure to unfavorable environmental conditions, a high concentration of this substance can cause a negative effect on physiological process courses [155]. An excessive melatonin amount can come from improper supplementation based on supraphysiological doses of melatonin or dysfunction of the organs responsible for the secretion of this hormone. This can cause circadian rhythm disorder, by imitating “artificial darkness” [156]. High concentrations of melatonin are associated with a high amount of its metabolites, which could have deleterious effects per se. Due to the knowledge of the pharmacodynamics of melatonin, the consequences of its high concentration may concern the signaling of the immune system, the central nervous system, platelet aggregation, and the cardiovascular system, as well as glucose metabolism, ending in carcinogenesis [157].

Pre-treatment with this indoleamine suppresses the increased intracellular level of ROS in TGF-β1-treated cells [158,159]. The antioxidant activity of melatonin can also attenuate epithelial-mesenchymal transition (EMT) stimulated by TGF-β1, by significant reversing changes in mRNA levels of *E*-cadherin, smooth muscle alpha-actin (α-SMA), vimentin, and fibronectin after TGF-β1 stimulation [160]. The TGF-β signaling pathway is mainly driven by a series of phosphorylation of Smad transducer proteins and their nuclear co-location to regulate the expression of target genes [161]. Melatonin prevents TGF-β1-induced cellular processes via the inhibition of Smad and non-Smad signaling cascades by inhibiting ROS-mediated mechanisms (Figure 1) [162]. Mechanistically, melatonin suppresses Smad2/3 phosphorylation and nuclear co-localization of their phosphorylated forms and Smad4 after TGF-β1 stimulation, in a dose-dependent manner [163,164]. The increasing phosphorylation of extracellular signal-regulated kinase 1/2 and p38 is attenuated by melatonin in a dose-dependent manner [165]. It is documented that the inhibitory action of melatonin does not require its membrane receptors [166]. The anti-inflammatory and anti-fibrotic actions of melatonin were also seen in the heart of melatonin-treated mice with diabetes mellitus, where it was found that melatonin significantly ameliorates cardiac dysfunction by inhibiting TGF-β1/Smad signaling and NOD-, LRR- and pyrin domain-containing protein 3 (NLRP3) inflammasome activation, as manifested by downregulating TGF-β1, *p*-Smad2, *p*-Smad3, NLRP3, ASC, cleaved caspase-1, mature IL-1*β*, and interleukin-18 (IL-18) [167].

## 5. Inflammatory EVs and Melatonin: Where Their Pathways Intersect

Different types of cells and tissues in the human body secrete distinct vesicles, which in turn allow for the transmission of intercellular signals through many pathways and, as a result, determine the maintenance of body homeostasis [168]. M137utual signal transduction via the membrane mediators is an important aspect of the body’s defense mechanisms because cells of the immune system are available to EVs for efficient and effective coordination of the immune response through the transport of biological molecules, primarily based on the proteins [169,170]. The course of immunological processes depends on the synchronization of a number of regulators of the immune system, both pro-inflammatory and acting as immune reaction brakes [171,172]. Among these biologically active pleiotropic compounds, significant importance has previously been ascribed to TGF-β, as well as the constitutive TNF-α molecule [173,174]. These substances mediate the secretion of membrane inflammatory mediators, affecting the frequency of their secretion, the number of secreted vesicles, and their molecular profile. Referring to the latest scientific reports, melatonin is also a factor that deserves special attention, as it can not only affect the mechanism of EV secretion but also modifies their cargo [175,176].

Melatonin is a widely acting anti-inflammatory molecule responsible for inhibiting chronic and acute inflammation, but it also removes ROS, which testifies to the antioxidant role of this compound [177,178]. The combination of properties characterized by EVs and melatonin-acting mechanisms seems to be a promising therapeutic strategy [179,180]. Accordingly, the melatonin-derived modification of the EVs cargo is considered a potential factor influencing damaged cells, noticeably modifying their molecular profile [181]. As a result, the presence of modified EVs in the environment of damaged cells can induce significant changes in the signaling mechanisms of these cells, often affecting their further fate [182].

### 5.1. Overview of Origin, Composition, and EVs Significance

EVs form a heterogeneous group of nanoparticles characterized by appropriate surface receptors (Table 1). Specific protein markers may be associated with different properties of the vesicles, affecting their ability to induce programmed death against different types of target cells or affecting their immune system stimulation [169,182,183,184]. Achieving the described relationships between cells through membrane vesicles secreted into the intercellular space enables an autocrine and paracrine means of intercellular communication, which favors the modification of both local and distant microenvironments [185]. The diverse load of EVs, apart from core proteins that reflect their origin and function, may also contain proteins phenotypically and physiologically identical to primary cells responsible for the secretion of specific vesicle populations, which means that they may provide important information about the pathological processes of the medical state of some individuals [186,187,188,189,190].

### 5.2. Melatonin-Dependent EVs

Due to the multifunctional nature of melatonin, it is considered one of the most important molecules providing hope in the dilemma concerning the connection of the unique diverse functions of EVs with their clinical application [191,192]. Melatonin is considered as the cornucopia among other neuromolecules, and its introduction into the selected populations of EVs induces modifications which have a strong protective effect on the surrounding cells (Figure 2). This is due to the ability of this biogenic amine to restore homeostasis in the body by stimulating the action of antioxidant enzymes, while having the ability to directly remove reactive oxygen and nitrogen species. This indicates the strong regulating properties of melatonin and its metabolites against the immune system, additionally showing a protective effect in diseases associated with oxidative stress [193,194,195].

It has been reported that the Toll-like receptor (TLR4)/NF-κB pathway connected with melatonin activity increases the anti-inflammatory possibilities of EVs via the stimulation of macrophage polarization. Melatonin-promoted EVs lead to the transformation of the M2 macrophage type by a phosphatase and tensin homolog deleted in the chromosome 10/Protein kinase B (PTEN)/PKB signaling pathway [196,197]. Melatonin-promoted EVs are characterized by the decreased exposition of vesicles signature cytokines, including IL-1*β*, IL-18, IL-6, and tumor necrosis factor-alpha (TNF-α), while an increase in the release of anti-inflammatory factors IL-10 and the conversion of TGF-β are observed [198,199].

Melatonin is a powerful antioxidant that scavenges various types of free radicals in body fluids and cells [200]. It has a protective effect against cellular oxidative stress, which includes anti-apoptotic actions. Melatonin-mediated EVs may play a neuroprotective role by upregulating the expression of the anti-apoptotic B-cell protein gene, which is observed in many neoplastic diseases, such as lymphomas [201,202]. The increased level of melatonin in the ECM environment modifies the biogenesis of EVs by regulating the secretion process from donor cells. Due to the relatively small size of EVs, their secretion may follow a mechanism characteristic of the secretion of low-molecular-weight metabolites, known in the literature as exocytosis. The mentioned process is based on the connection of secretory vesicles with the plasma membrane and the release of vesicle content into the extracellular space, leading to the selected proteins and lipids’ inclusion into the plasma membrane [203,204,205,206]. In the first step, melatonin activates the phosphatidylinositol 3-kinase/protein kinase B (PI3K/PKB) axis, while inhibiting the activity of glycogen synthase kinase 3 (GSK-3) [207]. During melatonin-supported exocytosis, an elevated donor cell membrane potential and increase in the elasticity and fluidity of the membrane have been observed. These symptoms are the result of the impending release of EV. There is also an increased metabolism of fatty acids in the cells responsible for extracellular secretion. Thus, the presence of melatonin in the cellular environment may result in enhanced intercellular communication, leading to increased exosome secretion [208].

Interestingly, the secretion of EVs can be supported by the process known as self-clearing the cells, described as autophagy [209]. Autophagy is considered as a regulated self-degradation process that can modify the mechanisms of exosome biogenesis in response to changes in external stimuli, related in this case to the presence of melatonin in the environment [210,211,212,213]. This is indicated primarily by reports which show that proteins responsible for the initiation and progression of cytosol and membrane autophagy take an active part in the formation and secretion of exosomes. Guo and Gil proved that the regulation of the autophagic system is based on the same signal transduction pathways as the formation of EVs [214]. The common denominator for these two, so far independent, processes, has become a conservative interaction involving the autophagy-related protein (ATG), such as ATG5 and ATG16L1 proteins [215]. Moreover, the presence of melatonin in the cellular environment directly induces autophagy by activating a number of proteins from the ATG protein family (4, 5, 7, 10, 12, 16), and the ratio of microtubule-associated protein 1A/1B-light chain (LC3) II/I is increased. In the context of EV secretion, the increased expression of these proteins enhances the fusion of multivesicular bodies (MVBs) with autophagic vacuoles and generates hybrid vesicles [216,217,218,219]. Due to the activity of specific guanosine triphosphatases (GTPases) such as Ras-related protein Rab-8A and Ras-related protein Rab-27 (Rab8a and Rab27), the release process of exosomes and autophagic contents is being carefully studied [220,221]. Thus, the presence of melatonin affects the overexpression of proteins responsible for the efficient course of autophagy, thereby changing the exosome production pathways and their content. Melatonin’s enrichment of the ECM may also provide the activation of other, alternative exosome secretory pathways, which are not observed in the melatonin-free environment [222,223].

The results of the bioinformatic analyses indicate a high correlation between the Wnt pathway proteins and melatonin-induced production of EVs. Based on the conducted analyses, the expression of paired box 2 (Pax2) and transducing-like cleavage enhancer 4 (TLE4) genes, characteristic of the course of the Wnt pathway, significantly enhances the secretion of EVs. Overexpression of these genes leads to the induction of specific intracellular signals, which then regulate the biogenesis of exosomes. However, the influence of the presence of melatonin on the alternative pathways of exosome biogenesis described above requires more extensive explanation in further studies [215,216,217,218,219,220,221,222,223].

The latest scientific reports indicate that the presence of melatonin in the environment of EVs can affect the size of exosomes, depending on the cells from which they were secreted [60,224]. The source of melatonin’s varied influence on the size of vesicles is therefore at the basis of the mechanisms of their biogenesis. For example, the presence of melatonin in bovine granulosa cells enhances the production of exosomes, but their diameter is much smaller than that of physiologically secreted exosomes. In some cases, i.e., SH-SY5Y human neuroblastoma cells, the presence of melatonin decreased the size of the vesicles, reducing it by as much as 36.23% [60,225]. Another feature of EVs influenced by the presence of melatonin in the ECM is their content, which is based primarily on mRNA, miRNAs and proteins [60]. In line with the topic of this work, the intercellular environment of anti-inflammatory macrophages enriched in melatonin induces an increase in the expression of transferring exosomes miRNA, such as miR-135b, miR-34a and miR-124, which distinguishes them from other EVs in which the above-mentioned tryptophan derivative is not observed [60,226]. Moreover, it has been shown that melatonin-stimulated EVs are characterized by increased levels of miR-4516. In the case of EVs secreted by smooth muscle, preliminary studies indicate that the presence of melatonin induces an increase in the expression of miR-204 and miR-211 in exosomes, compared to vesicles whose content was not subjected to melatonin modifications. The elevation in miR-181 in melatonin-treated exosomes is also observed in vesicles secreted by bone tissue cells, while enhancing the effect of osteogenesis. Notably, high expression of these molecules regulating the expression of exosome genes induced by the presence of melatonin may attenuate inflammation by modulating the immunoregulatory properties of EVs against target cells [60,225,226,227,228].

The presence of melatonin does not, however, affect all properties of exosomes. For example, high levels of melatonin in the cellular environment leave the unchanged tetraspanin levels of CD9, CD63, and CD81. The presence of melatonin also does not involve apoptosis regulators such as apoptosis-linked gene 2-interacting protein X (ALIX) and tumor progression genes, among which the tumor susceptibility gene 101 has been distinguished. The expression of protein markers on the surface of exosomes is extremely important in terms of the functionality of these structures. These specific surface molecules regulate internalization, immune evasion, and targeted exosome transport to target cells [60,225,226,227,228]. The issue of the level of proteins constituting the cargo of EVs, such as cellular prion protein (PrPC) or α-ketoglutarate, is different. Enrichment of the intercellular environment with melatonin affects the growth of these proteins inside the exosomes, which increases the proliferation and release of angiogenic cytokines to mitigate the inflammatory response [228,229,230].

In addition, exosomes are known to horizontally transfer melatonin between cells. Passive melatonin transport across the cell membrane is possible due to its lipophilic layer and passive membrane diffusion. Researchers are extremely interested in the mechanism of transfer of the high-affinity G protein-coupled receptors named MT1 melatonin receptor by internalization and endocytic transport. When melatonin signaling is connected to MT1, Rab5 supports the relocation of the internalized MT1 to early endosomes. MT1-carrying endosomes can cross the plasma membrane through the activity of other GTPases [60,220,221,222,223,224,225,226,227,228,229,230,231,232,233].

## 6. Conclusions

The development and complications of cardiovascular diseases are largely based on pro-inflammatory cytokine-signaling pathways. Dysfunction and activation of the endothelium leading to inflammation occur in response to the induction of ROS production by a myriad of proinflammatory cytokines. Due to its antioxidant properties, melatonin is a pharmaceutical that specifically targets the molecular and signaling pathways involved in the pathophysiology of CVDs, which has been demonstrated in the example of TGF-β. TGF-β is a cytokine that causes the growth and proliferation of many types of cells, and thus affects the increased production of EV populations. Despite the existence of many papers describing TGF-β-dependent vesicle secretion and melatonin-stimulated follicular secretion, there is still a lack of data confirming the synergy of these two factors in the secretion process.

## Figures and Tables

**Figure 1 metabolites-13-00575-f001:**
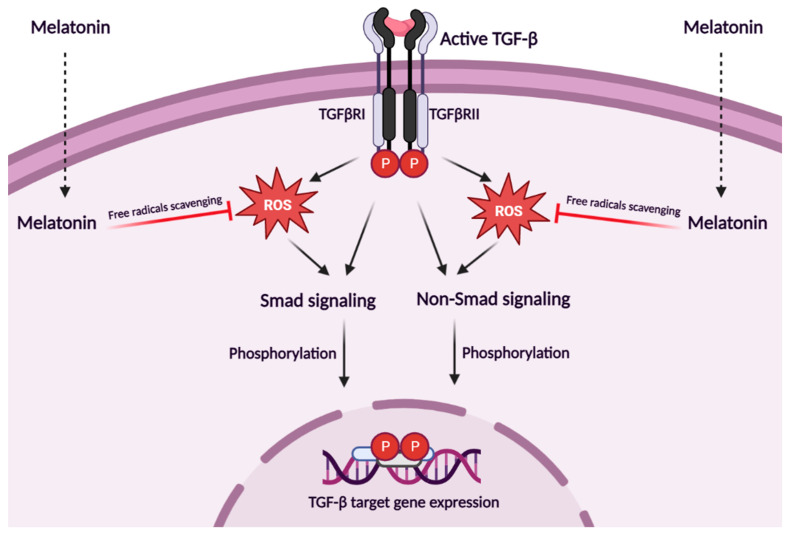
**Effect of melatonin on the TGF-β signaling.** TGF-β can signal via the canonical Smad proteins or in a Smads-independent manner. The specific course of the signaling pathway induced by the active TGF-β ligand depends on a series of phosphorylation of protein signal transducers. Intracellular levels of ROS are elevated after treatment with TGF-β1, while their presence ensures proper regulation of its signaling cascades. The hormone melatonin suppresses the TGF-β pathway due to its intracellular redox-status-altering properties, as evidenced by the effective reduction in ROS generation. Moreover, the inhibitory effect of melatonin is independent of its membrane receptor mechanisms. Indirectly, melatonin may interfere with many cellular processes coordinated by TGF-β-induced genes and intracellular ROS levels [158,159,160,161,162,163,164,165,166,167].

**Figure 2 metabolites-13-00575-f002:**
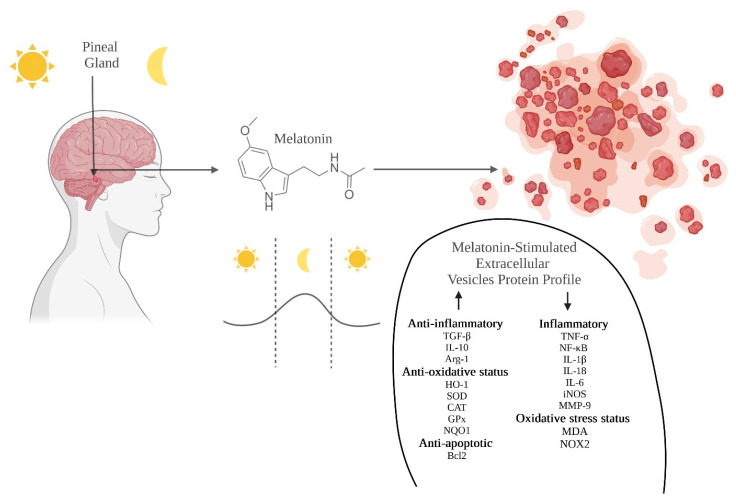
**Circadian rhythms in regulation of melatonin-dependent secretion of the EVs.** Melatonin is a ubiquitous molecule, synthesized in the pineal gland, and has myriad biological functions which primarily lead to the regulation of the endocrine circadian rhythm of the body. The presence of melatonin in the cellular environment changes the molecular composition of EVs. Melatonin is characterized by anti-inflammatory functions. The connection of EVs and melatonin represents a promising therapeutic instrument [191,192,193,194,195].

**Table 1 metabolites-13-00575-t001:** Detail classification and characteristics of EV populations (modified based on 186–190).

Vesicle Type						
	Exosomes	Microvesicles	Apoptotic Bodies	Oncosomes	Exophers	Migrasome
Morphology (by TEM)	Cup shape	Irregular shape	Oval shape	Heterogeneous	Quasi-Spherical Bodies	Pomegranate-like structures
Diameter (nm)	30–200	50–1000	50–5000	1000–10,000	+/−4000	500–2000
Density (g/mL)	1.13–1.19	1.04–1.07	1.16–1.28	N/A	N/A	N/A
Biogenesis	ESCRT endocytic pathway Ceramide-dependent multivesicular bodies	Cell Surface; Plasma membrane shedding	Cell Surface; Release by cell fragmentation during shrinkage caused to cell death (apoptosis)	Plasma membrane blebbing from cells	Budding out of cells into the extracellular space	Retraction fibers; Migracytosis
Enriched Markers	CD63 CD9 CD81 CD82 Hsp60 Hsp70 Hsp90 ALIX TSG101 PDCD6IP LAMP1 Flotillin-1 Rab27 ESCRT proteins	CD14 CD31 CD34 CD51 CD62E CD40 LL-37 HMGB1 ARF6 Integrin β1 VAMP3 ADAM10 NOTCH2	Trp-BODIPY cyclic peptide Annexin V C3b gp96 PANX1 Caspase-3 Caspase-7 VDAC1	CD63 CD9 CD81 Cytokeratin-18 EGFR AKT1 Cav-1 ARF6 CK18 MMP-2 MMP-9 eEF1γ αV-integrin MDH GPI-Aps	MAP2 β-III tubulin tau protein	Tspan-4 Tspan-7 Integrinα5β1 NDST1
Molecular Cargo	Lipid Proteins Nucleic acids Non-coding RNAs MHC molecules	Lipid Proteins Nucleic acids Non-coding RNAs	Nuclear fractions	Protein Nucleic acids Non-coding RNAs	Cell organelles	
Processes	Intercellular communication via paracrine, autocrine, endocrine, and cell-to-cell contact signaling					
Detection	Flow Cytometry ELISA Cryo-EM TEM SEM WB AFM DLS RPS Proteomics			TEM SEM IF WB		

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
