# Peer review of "Melatonin and TGF-β-Mediated Release of Extracellular Vesicles"

_metabolites, 2023, doi:10.3390/metabo13040575_

Round 1

Reviewer 1 Report

The review “Melatonin and TGF-β-Mediated Release of Extracellular Vesicles” by Piekarska et al. is of potential interest for a wide readership. However, improvements of the writing style and content are necessary to make this manuscript ready for submission.

Please carefully check the text for typos and errors of grammar. Many statements are rather vague. The manuscript can improved by rephrasing sentences especially in the Introduction.

Some of the structure of the manuscript is strange. For example, section 4 is entitled “Effect of Melatonin on the TGF-β Signalling”, but the first paragraph of this section does not at all mention melatonin.

Figure 1 leaves many important questions open. How does melatonin block ROS?

Line 265: Please rephrase “Intracellular ROS are … a product of TGF-β”. This would sound like TGF-b generates ROS in a kind of enzymatic reaction.

Figure 2 can also be improved by adding mechanistic data.

Table 1: Which forms of caspase-3 and caspase-7 are markers of apoptotic bodies? Probably, only the active/mature forms of these proteins are specific for apoptosis. Is it true that nuclear fractions are the only “cargo” of apoptotic bodies?

Author Response

Reviewer #1:

Point 1: The review “Melatonin and TGF-β-Mediated Release of Extracellular Vesicles” by Piekarska et al. is of potential interest for a wide readership. However, improvements of the writing style and content are necessary to make this manuscript ready for submission. Please carefully check the text for typos and errors of grammar. Many statements are rather vague. The manuscript can improved by rephrasing sentences especially in the Introduction.

  • Authors’ response: We sincerely appreciate the reviewer’s comments. We have reformulated and revised the manuscript accordingly, additionally supported by the native speaker.

Point 2: Some of the structure of the manuscript is strange. For example, section 4 is entitled “Effect of Melatonin on the TGF-β Signalling”, but the first paragraph of this section does not at all mention melatonin.

  • Authors’ response: Thank you very much for this valuable remark. We agree with the reviewer, and we have added a paragraph describing melatonin and its properties.

Point 3: Figure 1 leaves many important questions open. How does melatonin block ROS? Line 265: Please rephrase “Intracellular ROS are … a product of TGF-β”. This would sound like TGF-b generates ROS in a kind of enzymatic reaction.

  • Authors’ response: Thank you for these suggestions. We referred to both remarks, extending the content about free radicals forms functions in the melatonin-dependent TGF-β signaling pathway.

Point 4: Figure 2 can also be improved by adding mechanistic data.

  • Authors’ response: Thank you very much for this suggestion. It would have been interesting to extend this aspect. In our humble opinion, this would not be possible in our review because the mainstream of the text focuses on the secretion of extracellular vesicles dictated by the regulation of the TGF-β signaling pathway connected with melatonin activity. Respecfully to the reviewer’s comment, this review focuses not only on the on the melatonin's mechanism of action. Herein, we do agree that the development of mentioned issue by the reviewer would be an excellent topic for further discussion and work.
  •  

Point 5: Table 1: Which forms of caspase-3 and caspase-7 are markers of apoptotic bodies? Probably, only the active/mature forms of these proteins are specific for apoptosis. Is it true that nuclear fractions are the only “cargo” of apoptotic bodies?

  • Authors’ response: We thank for pointing this out and we admit that this is an excellent suggestion. As mentioned by earlier reports, mature caspases-3 and -7 cleave a large set of substrates, ultimately resulting in the characteristic morphological and biochemical hallmarks of cells under the apoptotic progression. However, the table contains overall information on enriched markers and does not differentiate them according to the level of development of selected forms of vesicles. On the other hand, utilizing a flow cytometry-based approach, it has been demonstrated that intracellular contents including nuclear materials are distributed to some apoptotic bodies.

Reviewer 2 Report

The authors have submitted a review article of illustrating a current knowledge regarding impact of physiologically released melatonin on transforming growth factor-beta (TGF-beta)-mediated release of extracellular vesicles via mechanisms dependent or independent of intracellular protein SMAD-induce signals in humans under the activated immune system. The authors searched a range of eligible literature, from well-known classical, and latest research regarding an association of melatonin with activation of inflammatory signaling pathways, which are primarily attributed to the flexibility of the immune systema and as a result homeostasis in human body. It is of interest that the authors discussed the beneficial availability of melatonin and the pharmacologic properties which regulates the states of inflammatory responses, resulting in reliable perspectives. This issue is of quite interest, and impact of their review is strong. My overall concern with the review describing the current available data regarding beneficial availability of melatonin against regulation of the immune system is that information provided may offer something substantial that helps advance our understanding of effective management which draws novel class of effective medicines towards inflammatory signaling pathways available in clinic. The reference list may be useful for readers who are interested in this issue.

To strengthen authors’ perspectives, the authors are strongly recommended to add a “toxicology” sub-section regarding known melatonin side-effects on humans, for instance. The opposite, toxicological effects of expected outcomes, if known, may influence largely the authors’ perspective for the effectiveness of melatonin. In addition, the authors are requested to add the information regarding the transport mechanism(s) of melatonin from extracellular space into the intracellular domain because melatonin are expected to exert its effect on the immune systems independent of its own cell-surface receptors.

Author Response

The authors have submitted a review article of illustrating a current knowledge regarding impact of physiologically released melatonin on transforming growth factor-beta (TGF-beta)-mediated release of extracellular vesicles via mechanisms dependent or independent of intracellular protein SMAD-induce signals in humans under the activated immune system. The authors searched a range of eligible literature, from well-known classical, and latest research regarding an association of melatonin with activation of inflammatory signaling pathways, which are primarily attributed to the flexibility of the immune systema and as a result homeostasis in human body. It is of interest that the authors discussed the beneficial availability of melatonin and the pharmacologic properties which regulates the states of inflammatory responses, resulting in reliable perspectives. This issue is of quite interest, and impact of their review is strong. My overall concern with the review describing the current available data regarding beneficial availability of melatonin against regulation of the immune system is that information provided may offer something substantial that helps advance our understanding of effective management which draws novel class of effective medicines towards inflammatory signaling pathways available in clinic. The reference list may be useful for readers who are interested in this issue.

Point 1: To strengthen authors’ perspectives, the authors are strongly recommended to add a “toxicology” sub-section regarding known melatonin side-effects on humans, for instance. The opposite, toxicological effects of expected outcomes, if known, may influence largely the authors’ perspective for the effectiveness of melatonin. In addition, the authors are requested to add the information regarding the transport mechanism(s) of melatonin from extracellular space into the intracellular domain because melatonin are expected to exert its effect on the immune systems independent of its own cell-surface receptors.

  • Authors’ response: We appreciate very much the reviewer’s feedback. We are grateful for the valuable comments that certainly lead to improvements of this report. The content of our manuscript has been enriched with suggested information, thanks to which the work reflects, even a synthetic overview of information on the structure, properties, and functions of melatonin. Unfortunately, the literature does not indicate the direct, toxic effects of melatonin, supported by a number of scientific studies, but in the case of high concentrations of this hormone, the circadian rhythm disorders should be expected. Moreover, the review was enriched with informations about melatonin‘s activity that does not require interactions with the melatonin receptors located in the other cells, highly involving the mitochondrial functions. The continuation of this review could be independent report containing a comprehensive description of melatonin activity and its transport inside cellular compartments, based on the GLUT/SLC2A and PEPT1/2 transport system.

Round 2

Reviewer 1 Report

The revised manuscript contains a better description of melatonin.

Reviewer 2 Report

The authors have done a good job responding to reviewer comments and concerns in their revision. I believe the manuscript is improved as a result. Now I recommend that this revised version of the manuscript can be accepted for publication in the journal Metabolites.